# Quality of Eggs and Albumen Technological Properties as Affected by *Hermetia Illucens* Larvae Meal in Hens’ Diet and Hen Age

**DOI:** 10.3390/ani10010081

**Published:** 2020-01-03

**Authors:** Giulia Secci, Fulvia Bovera, Giuliana Parisi, Giuseppe Moniello

**Affiliations:** 1Department of Agriculture, Food, Environment and Forestry, University of Firenze, via delle Cascine 5, 50144 Firenze, Italy; giulia.secci@unifi.it; 2Department of Veterinary Medicine and Animal Production, University of Napoli Federico II, via Federico Delpino 1, 80137 Napoli, Italy; fulvia.bovera@unina.it; 3Department of Veterinary Medicine, University of Sassari, via Vienna 2, 07100 Sassari, Italy; moniello@uniss.it

**Keywords:** black soldier fly, foam capacity, angel cake, texture, week of deposition

## Abstract

**Simple Summary:**

Concurrently with the actual challenges in the poultry industry, we aimed to assess the changes induced by the inclusion of an alternative protein source (partially defatted *Hermetia illucens* larva meal, HI) at two different levels as well as hen age on the overall egg quality and deepened their effect of albumen technological properties. This study could provide useful information for the egg supply chain in order to optimize egg utilization, as a whole or as egg products, thus preventing food losses or undesirable wastes. Specifically, based on the obtained results, the eggs laid by hens fed the highest inclusion of HI would be directed towards egg product industry due to their reduced eggshell percentage and thickness which could increase their fragility. Contrariwise, due to the impaired albumen technological properties, as foaming, found in the egg laid by 27–35 wk-old hens, they could be preferentially sold as a whole.

**Abstract:**

The impact on the egg quality and the albumen technological properties were here evaluated as affected by diet and hen age (A) of 162 Hy-line Brown hens. Three isoproteic and isoenergetic diets were formulated respecting the requirements for Hy-line hens: the control diet (C) based on conventional protein sources, and other two where vegetable proteins were substituted at 25% (HI25) and 50% (HI50) by partially defatted *Hermetia illucens* larva meal (HI). Ten eggs collected from each group at the hen ages of 20, 27, and 35 weeks were evaluated. The eggshell percentage and thickness were significantly reduced in the HI50 eggs (11.93% and 476 µm, respectively) compared to the C (12.34%, 542 µm) and HI25 (12.54%, 516 µm). The aging lowered (*p* = 0.05) the protein and increased (*p* < 0.001) water contents of the eggs. Although the foam capacity of the HI50 albumen was halved than the C group (*p* < 0.05), it was unaffected by the aging. Additionally, this did not impair the volume and the textural properties of a batter (angel cake) in which it was included. On the opposite, the textural characteristics of the cake made by the oldest hens (i.e., 35 wk-old) were compromised. In conclusion, the diet and hen age differently affected egg quality and its technological properties, which could be positive to obtain eggs to destine directly to the market or to the egg industry.

## 1. Introduction

Standing on the data proposed by the Food and Agriculture Organization in 2018 [1], eggs are the second-fastest growing industry in the world, with more than 50% growth forecast in the next 2 decades. Europe production in 2019 amounts to 7.5 million tonnes of eggs thanks to the 400 million laying hens kept throughout the European Union (27 Member States) [2]. Italy is the fourth producer with approximately 817,000 tons of eggs consumed as a whole or destined to the egg industry. Indeed, despite eggs play a very important role in human nutrition as a precious source of proteins, essential amino acids, lipid and several trace elements, the industry is continuously asking for egg products, such as liquid, powdered, concentrated whole egg or its separated main components, yolk and albumen. For instance, the egg allowance directed toward processing chains has been estimated at 32% of the whole Italian production [3]. This interest is linked to the unique functional properties of eggs, such as foaming, gelling and emulsification. The foaming capacity belongs to the egg white protein and it is defined as “the ability to rapidly adsorb on the air-liquid interface during whipping or bubbling and by its ability to form a cohesive viscoelastic film by way of intermolecular interactions” [4]. Thanks to these characteristics, albumen is widely utilized as an ingredient in bakery products, like bread, cakes and meringues, ice creams and several other processed foods.

The increase in egg and egg-derived product demand is however dependent on the laying hen farming, a sector that is affected by some critical aspects, such as feed supplying. As one of the main concerns of the last decade, the unsustainability of feed, especially soybean meal, is quite debated. Indeed, despite being an important source of essential amino acids, soybean production for feed directly competes with human nutrition and is largely associated with water pollution and exploitation of the lands [5].

A big opportunity for companies looking at more sustainable protein sources has been recently identifying in insect meals. Although processed animal proteins (PAPs) are still banned as livestock feeds in Europe, the European Commission voted in 2017 to introduce seven insect species, processed as meal or oil, into fish feeds and pet food [6]. That is why the stakeholders are confident that these protein sources would be allowed in other feeds in the coming years. As the interest in this alternative ingredient has grown, researchers are checking the feasibility of partial or complete substitution of the conventional protein sources with insect meal in poultry feed. Among the other species, *Hermetia illucens* meal (HI) has deserved major attention for laying hen diets. Indeed, in addition to the other advantages related to the rearing process of this species of insect, HI is a valuable protein source (40–44 g/100 g of dry matter, DM) and it provides calcium and phosphorous (50 to 80 g/kg dry matter, 6 to 15 g/kg dry matter, respectively), which are fundamental for laying nutrition [7]. HI has been proved to be a feasible substitute for vegetable protein sources in the hen diet when considering animal growth and performance [8,9,10] and egg quality [11,12,13]. However, the authors focused on the egg deposition, quality characteristics as the albumen height and the yolk color and its chemical composition, more than the albumen technological properties, which could be of interest to the egg industry. Thus, this study aimed to test the effect of partial substitution of the conventional protein sources with a partially defatted *Hermetia illucens* larva meal on the overall egg quality and on the technological properties of the albumen from the eggs produced by Hy-line Brown laying hens and collected at three different hen ages.

## 2. Materials and Methods

### 2.1. In Vivo Trial

All the hens were treated in accordance with the Directive of the European Parliament of the Council on the Protection of Animals Used for Scientific Purpose and in agreement with the Institutional Animal Care and Use Committee of the University of Naples Federico II, D.lgs n. 26 04/03/2014. All the experiments involving hens were approved by the Bioethical Committee of the University of Naples Federico II, under the number of protocol 2017/0017676.

The HI tested in this study was purchased from Hermetia Deutschland GmbH & Co KG (Baruth/Mark, Germany), chemically characterized as reported in Bovera et al. [14] and utilized to formulate the experimental diets, presented as follows. A control diet was formulated based on the conventional protein sources (corn-soybean based diet, C) while the other two were based on two different substitution levels of the conventional protein source with the partially defatted *Hermetia illucens* larva meal (HI, ether extract equal to 8.34% as fed). Specifically, the 25 and 50% of the dietary proteins were substituted by HI in the H25 and HI50 groups, respectively. The diets, whose ingredients and chemical composition are shown in Table 1, were formulated in order to result isoproteic and isoenergetic and to satisfy the Hy-line Brown requirements, as specified in Bovera et al. [14]. A total of 162 sixteen weeks-old Hy-line Brown hens (average live weight 1.41 kg ± 0.13) were housed in a commercial private laying hens farm (total flocks housed: 40,000 hens), located in Sardinia (Italy), for 20 weeks. The hens were equally divided into three feeding groups. Each group was housed in modified cages (18 hens/cage, 800 cm^2^/hen, nine replicates of six hens each) adjacent to a nest box for egg deposition. The trial started after four weeks of adaptation, when hens were 20 weeks old. The mashed feeds were administrated *ad libitum* to the hens through a feeding belt, manually loaded. A dark: light cycle of 9:15 h was set. More details concerning the farming conditions as well as the data on feed intake, feed conversion ratio, weight gains and laying percentage can be retrieved in Bovera et al. [14]. The collection of the eggs for the purposes of the present trial occurred at three different hen ages (A), i.e., namely 20-wk-old (A20), 27-wk-old (A27), and 35-wk-old (A35). Specifically, the eggs were collected every day, then the overall weekly production per each dietary treatment was pooled. Ten undamaged eggs were selected from each pool and destined to the present trial, hence stored at −80 °C and moved to the Department of Agriculture, Food, Environment and Forestry (University of Firenze, Italy) where they were subjected to the scheduled analyses.

### 2.2. Egg Quality and Albumen Foaming Properties

After an over-night thawing in a refrigerated room (4 °C), the eggs were weighed prior to being broken and to separate each component (eggshell, albumen, and yolk) one from each other. The weights of the albumen and yolk were recorded and utilized for the calculation of their relative percentage. Concerning the eggshells, they were weighed after the removal of the inner membrane and the drying of the internal surface with common kitchen paper. After this, the thickness of the eggshell was measured by a manual caliper (Salmoiraghi, Milan, Italy) on three points (equator, round and apex). After this, the whole eggshell was grounded prior to ash quantification [15]. The color of the yolk was measured on three points by a Konica Minolta colorimeter CR-200 (Chiyoda, Japan) and the data were expressed as lightness (L*), redness (a*) and yellowness (b*) indexes [16].

The following analyses were conducted one-by-one on each albumen, without pooling the ones belonging to the same treatments. The chalazae were removed from each albumen prior to gently homogenate it by stirring for 15 min (speed set n. 4 of the magnetic stirrer, Type Age, Velp Scientific, Usmate Velate, Italy) in order to avoid foam formation and water losses, then the pH was determined using a SevenGo pH meter (Mettler-Toledo, Columbus, OH, USA) at room temperature. Ten mL of each albumen were allotted to the evaluation of egg white technological properties, as foam ability and foam stability, following the methodology recently proposed [17]. Briefly, the albumen was whipped for 1 min at 20 °C in a 50 mL plastic cylinder by a high-speed mixer (IKA Werke T25, Staufen, Germany) operating at 9500 rpm and the volume of the foam was recorded 30 s after whipping (V0) and holding 30 min (V30). The data were inserted into the following formula for the determination of the foam capacity (FC, %) and the foam stability (FS, %) [18]:FC = (V0/Valb) × 100(1)
FS = (V30/V0) × 100(2)
where Valb is the initial volume of the albumen.

The other 20 g of the egg white from each sample were destined to the angel cake preparation, as detailed in the next section. The remaining albumen (around 8 g) was freeze-dried prior to being subject to water and crude protein content quantification [15].

### 2.3. Angel Cake Preparation and Characterisation

Twenty g of egg white, 14 g of white sugar (Italia Zuccheri, Minerbio, Italy), 8 g of wheat flour “00” (Il Molino Chiavazza, Casalgrasso, Italy), and 2 g of rice starch (Pedon S.p.A., Molvena, Italy) were utilized for the production of the angel cake (one for each sample), which is a common batter utilized to test the foaming capacity of the protein ingredients. Albumen and sugar were whipped together for 60 s with a hand mixer equipped with a wire whisk attachment (Kenwood HDP408WH Triblade Mixer, Kenwood, Woking, UK), set at its maximum speed level. All the other dry ingredients were manually incorporated into the foam by beating on high until stiff peaks form (approximately 40 s). An aliquot corresponding to 31.711 ± 1.072 g of batter was spooned into one cavity of an ungreased silicone pan (mold size: 23.5 × 21.2 × 2.5 cm, total 6 cavities, cavity size: 8.0 × 5.5 × 2.5 cm, Shen zhen shi jun yang ke ji you xian gong si, Guangdong, China). The batters were baked for 15 min in a pre-heated static oven (180 °C, Mod. S370EB, Smeg, Guastalla, Italy), then the angel cakes were left to cool at room temperature for 1 h. Some physical parameters such as the weight and the maximum height of each cake were recorded, then baking loss (BL, %) was calculated as proposed [19]:BL = (WCk–Wb) × 100/WCb(3)
where WCk was the weight of the angel cake, Wb was the weight of the batter and WCb was the water content of the batter obtained following the AOAC method [15]. Both angel cake dough and the baked cakes were analyzed for water and protein contents [15].

The color and texture profile analyses were assessed. Specifically, the cakes were cut vertically in the middle. Once opened, the color values were determined with a Konica Minolta colorimeter (Chiyoda, Japan) [16] on two points of the two halves. Finally, 4 × 4 cm pieces were cut from the two halves of each cake, the crust was removed and then both pieces were analyzed, using a Zwick Roell^®^ texturometer (model KAF-TC 0901279, Zwick GmbH & Co., Ulm, Germany) equipped with a 50 N load cell. The texture profiles analysis was determined through a double compression cycle (crosshead speed: 1 mm s^−1^) until the 60% deformation of the initial height, as previously done [19]. The hardness (N), cohesiveness (the area of work during the second compression divided by the area of work during the first compression), springiness (the distance of the detected height during the second compression divided by the original compression distance) and chewiness (Hardness x Cohesiveness x Springiness) were calculated from the force–time diagram (Test-Xpert2 software version 3.0, Zwick GmbH & Co., Ulm, Germany) [20].

### 2.4. Statistical Analysis

The data collected were processed by a 2-way ANOVA using the PROC GLM of SAS/STAT Software, Version 9 [21] by using Diet (D), Hen Age (A) and their interaction (D × A) as fixed factors. The comparison among the means was achieved by conducting Tukey’s test [21] and the significance was set at *p* < 0.05. The means are accompanied by the Root Mean Square Error (RMSE) for characterizing the variability of the groups. Each egg within the feeding group and hen age was considered as a biological replicate.

## 3. Results

The egg, eggshell and yolk weights significantly increased due to the inclusion of HI in the feed for laying hens, irrespective of the insect inclusion level utilized (Table 2). Despite this, while looking at the percentage of the main components of the egg, only the eggshell resulted significantly diminished in the HI50 eggs compared to the C and HI25 groups. Accordingly, the thickness of the eggshell was reduced while increasing HI in the feed.

Furthermore, the eggshell percentage varied (*p* < 0.05) along with the hen age, showing the highest value at A27. These modifications in the eggshell characteristics cannot be attributed to the different ash content since it resulted unaffected by both D and A and equal to 76.4 ± 2.5 g/100 g of eggshell. Since L*, a*, and b* values resulted significantly affected by the interaction D × A at *p* < 0.026, 0.0001, and 0.049, respectively, we summarised the means obtained for this interaction in Table 3. No difference in the lightness of yolks was found as affected by the diet in the A20 group (Table 3). Conversely, the H50 eggs laid at A27 and A35 showed L* value significantly different from the C ones. The major differences were in the a* value, found significantly higher in the C yolks than in the H25 and H50 ones at A20 and A27. A greater decrease of the a* value was recorded in the C group increasing the time of laid, whereas the H50 raised its maximum a* value in the yolk from the eggs collected at A35. The b* index appeared scarcely affected by the diet, indeed the H50 yolks were significantly less yellow than the other two groups only in the A20 group. Furthermore, the same group significantly increased the b* index from the A27.

Table 4 shows the results of the technological properties and chemical composition analyses of the albumen. The FC was decreased by the presence of HI in the diet of laying hens, whereas the FS ad the pH value, as well as the water and protein contents were unaffected. A deep effect of the A can be observed for pH and chemical components of the albumen. For instance, a significant decrease in the pH value (*p* < 0.01) was found after the A27. The albumen of the eggs collected at the A35 showed a significantly (*p* = 0.05) lower protein content than those collected at the A20, while their water content increased with the aging.

Data related to the significant interaction between D and A emerged for the FC are reported in Figure 1. Although the H50 albumens had a halved (*p* < 0.05) FC than the C ones, they kept this property constant while increasing the hen age, while both the C and H25 groups reduced their FC between the A20 and the A35.

The FC of the egg white was even determined by the analyses of physical parameters conducted on the angel cakes (Table 5). The textural properties of the angel cake were not affected by the dietary inclusion of *Hermetia illucens*, while A20 induced a significant reduction of the chewiness.

A significant interaction (D × A) emerged for height, b* value and baking loss of the samples (Table 6). The height of the angel cakes produced from the C egg whites was unaffected by the A, whereas the angel cakes obtained by the H25 eggs collected during the A35 significantly lost their height compared to those produced from the H25 eggs of the A20. While looking at the difference among the dietary treatments, the angel cake from the C eggs was higher than the H50 only considering the A20. In addition, the H25 egg whites collected at the A27 and A35 significantly decreased the height of the angel cake compared both to the C and H50 groups. Concerning the baking loss, the age did not affect the water retention ability of the angel cake made with the C albumen, while the highest inclusion of HI in the diet increased the ability of the batter to retain water, especially at the A35. The b* value of the angel cake obtained with the C albumens was unaffected by the age, while the yellowness was significantly higher using the albumen from the H25 eggs collected at the A35 than that collected at the A20. Conversely, the H50 angel cake was slight but significantly discoloured when prepared with the egg whites from the A27 and A35. Regarding the differences among the diets, the angel cakes produced with the H25 albumens collected at the A20 had a lower (*p* < 0.001) b* value than those of the C and H50 groups. In contrast, the H50 cakes prepared with the eggs collected at the A27 and A35 showed a lower (*p* < 0.001) b* index than the C and HI25.

Finally, Table 7 depicts that the batters prepared with the egg whites collected at the A27 had a slight but significantly lower water content than those from the egg whites collected at the other two considered ages.

## 4. Discussion

The main factors affecting the egg weight are the dietary metabolized energy [22] and the size of the yolk that, in turn, is influenced by the body weight of the hens [23]. Since the administered diets in the present trial were isoenergetic, it seems clear to attribute the major changes in HI egg weights to the increase in the size of their yolks. Nevertheless, this modification cannot be attributable to the different body weight of the hens, resulted unaffected by the dietary intervention as well as the feeding intake and the feed conversion ratio [14]. Thus, the diet could have influenced the egg and yolk weights. Leeson and Summers [23] suggested that egg weight is very sensitive to methionine and total sulphur amino acid levels, which however were balanced in all the three administered diets. Our hypothesis is that the significantly (*p* = 0.032) augmented length of the jejunum found in the HI25 and HI50 hens, being respectively 4.68 and 4.64 (as percentage of the live weight), compared to the C one (3.45% live weight), [14] could have promoted the absorption of the amino acids, hence resulting in a higher assumption of these nutrients. This possible explanation strongly needs the support of other comprehensive studies, since the effect of HI meal on hen egg weight is still scarcely investigated and conflicting in results [13,24]. The increased egg weight found with the HI meal inclusion in the hens’ diet may require an increase in eggshell weight because of a rising in calcium deposition. Notwithstanding, authors underlined that egg-laying birds have a limited amount of calcium available to produce the shell, approximately 2.0–2.5 g Ca^2+^, irrespective of the egg size weight [25]. Hence, the production of heavier eggs in the HI50 group might explain the significant reduction in shell percentage and even in its thickness. Since the eggshell weight and thickness are physical variables correlated with the egg strength, resistance to physical and pathogenic challenges from laying to the transportation and selling phase [26], our results should be considered when evaluating the suitability to include the HI meal in the laying hens’ feed.

The egg internal quality, determined by several parameters (as albumen height and Haugh Unit), is even associated with the yolk quality, whose color is a fundamental characteristic [27]. The pigments contained in feeds mainly derive by plant ingredients, however, Secci et al. [13] underlined that even the HI larva meal may contain carotenoids and tocopherols. These pigments have a strong affinity for non-polar molecules, such as lipid, and they generally absorb wavelengths ranging from 400 to 550 nm, coloring as yellow, orange, or red. Thus, carotenoids can be easily accumulated in the egg yolk, which is rich in fat, producing the major modification of the yolk redness index, as we found.

Although the egg production and quality as affected by *H. illucens* in laying hen diets have been recently examined [11,12,13,24], the focus areas were the egg deposition and quality characteristics as the albumen height, the yolk color and its chemical composition. Nonetheless, the albumen has relevant importance for the egg industry mainly because of its unique functional properties, such as foaming that promote its extensive use as an ingredient in several processed food. In the present study, the diet did not affect the egg white pH and protein content, thus supporting previous findings [13]. The protein concentration as well as pH and cooking temperature, in addition to the physical-chemical properties of proteins, are well-established factors affecting the foaming ability [28]. Although in the present study no significant difference among the diets was found for pH and protein values, the foam ability of the HI25 and HI50 egg whites was decreased compared to that of the C ones. Since the temperature was controlled during the experiment, we hypothesize that the HI diets could modify the concentration, or the proportion of the single protein fractions contained in the egg white. As previously noted [4], the albumen foaming properties are the result of the interaction among the different proteins, such as globulins, ovalbumin, ovotransferrin, lysozyme, ovomucoid and ovomucin. Each fraction has its own ability to form a voluminous foam (foam capacity), to maintain it (foam stability) and to increase the volume of the batter containing it (i.e., angel cake) which differs from the overall foaming ability of the whole albumen [4]. For instance, globulins have the highest foaming index, around 4.71 cm^3^/g min, more than 7 times higher than that of the ovalbumin (0.59), while ovomucin and ovomucoid show no foaming ability [4]. Nevertheless, Mine [4] reviewed that these last two proteins contributed to the final volume of the angel cakes made with them, whereas the ovalbumin produced cakes with a volume comparable to the one obtained by globulins, being 308 and 330 cm^3^, respectively [4]. Standing on the mentioned literature, the possible role of the diet on protein matrix composition should be investigated to support or not our results about the technological properties of the egg white.

The albumen chemical composition varied while increasing the hen age, in line with previous findings [29] which proposed that the increase in hen age and weight led to a reduction in the crude protein content of the albumen. Changes in pH and protein content are consistent with the reduction of the foam capacity while using eggs from the 35th week of age. Instead, the maximum height raised by the angel cake varied due to the interaction D × WDA, thus supporting the previous hypothesis on the induced modification of proteins.

Baking loss is an important technological characteristic of a batter and plays a key role both for the quality of the final baked products and their shelf life. Indeed, the water content can promote microbiological growth during the storage, inducing a loss of the shelf life length. Considering that proteins may affect the water retention during cooking, the high affinity to the water of the white from eggs laid by hens fed the HI deserves other studies.

It has been previously demonstrated that the HI inclusion in the diet affects the yolk color values of the eggs [11,12,13], probably because of the pigments contained in the insect meal [13]. Standing on our knowledge, this is the first time that a difference in color emerged in a baked product made by egg white derived by hens fed with HI and, for this reason, little comparisons are possible.

The textural attributes of food are strictly related to its quality and consumers’ acceptance. In the present study, the HI meal presence in the diet did not affect the textural properties of the angel cake, similarly to other authors who found that different dietary types of protein (i.e., soybean meal, cottonseed protein, double-zero rapeseed meal, individually or in combination with equal crude protein) administered to the Jinghong laying hens did not affect the cooked yolk hardness and springiness [30]. Furthermore, the deep reduction in the chewiness that occurred between the eggs laid at 20 and 35 weeks of hens’ age agrees with the finite effect of the substitution of egg white by vegetable protein on angel cake chewiness [19]. Since chewiness can be enhanced due to a reinforcement of the protein entanglement in the networks [19], it is possible that variations in the egg white protein-gluten-starch connections occurred with hen aging, although more studies are necessary to define what kind of interaction is eventually developed.

## 5. Conclusions

The challenge for the foreseeable future will be to increase the sustainability of poultry production, including the egg supply chain. If one of the steps could be the substitution of the conventional protein sources, i.e., soybean meal, with the alternative ingredients, as insects in the case of their approval by European legislation, another point could be how to optimize the use of poultry products. In this regard, the present study provides the first information on how *infra-vitam* factors, such as diet and hens’ age, could affect egg components and albumen technological properties, thus suggesting a differential use of the eggs based on hens’ farming. Here, the use of eggs laid by hens fed the HI50 diet could be directed towards to egg product industry instead of selling as whole eggs because of their reduced eggshell percentage and thickness. Contrariwise, the eggs laid by hens from 27th to 35th weeks of age could be sold as whole because of their impaired albumen technological properties and consequently the chewiness of the baked cakes, which could be negatively perceived by the consumers.

## Figures and Tables

**Figure 1 animals-10-00081-f001:**
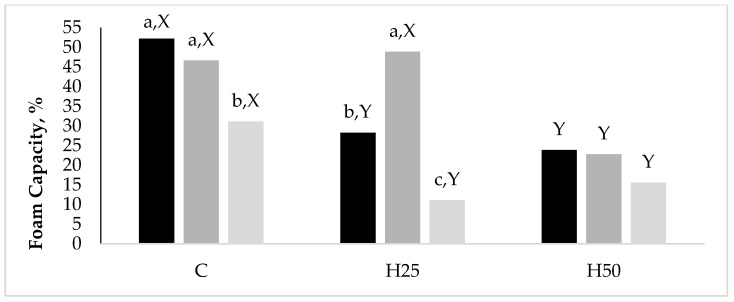
Effect of the interaction Diet x Hen Age on the foam capacity (%) of albumen from eggs laid by hens fed three experimental diets (C, H25, and H50) at three different hen ages (A20, black bars; A27, dark grey bars; A35, grey bars). a, b, c indicate significantly different means among the hen ages, within the diets. X, Y indicate significantly different means among the diets, within the hen ages.

**Table 1 animals-10-00081-t001:** Ingredients (g/kg), chemical composition (g/100 g as fed), and metabolizable energy (MJ/kg) of the experimental diets C (control diet based on maize grain and soybean meal), HI25 (*Hermetia illucens* larva meal substituted the 25% of dietary vegetable protein) and HI50 (*Hermetia illucens* larva meal substituted the 50% of dietary vegetable protein).

	Diets		
	C	HI25	HI50
Ingredients			
Maize grain	605.5	597.5	630.5
Soybean meal	265	200	95
Insect meal	-	73	145
CaCO_3_ grains	80	80	80
Vegetable oil	10	10	10
MinVit *	10	10	10
Methionine	2.5	2.5	2.5
Monocalcium phosphate	5	5	5
Celite	20	20	20
Salt	2	2	2
**Chemical Composition**			
Dry matter ^1^	91.53	91.39	91.62
Crude protein ^1^	16.45	16.32	17.03
Ether extract ^1^	3.17	3.61	4.06
Metabolizable Energy ^2^	11.85	11.90	11.89
Ash	14.00	14.21	14.13
Ca	4.63	4.75	4.51
Total P	0.68	0.69	0.65
Lysine ^2^	0.86	0.97	1.00
Methionine ^2^	0.63	0.58	0.61
Methionine + Cysteine	0.84	0.80	0.82
Isoleucine	0.72	0.73	0.75
Threonine	0.63	0.64	0.65
Tryptophan	0.18	0.15	0.16
Valine	0.93	1.07	1.14

^1^ Determined according to AOAC (2004). ^2^ Calculated according to NRC (1994).* Provided per kilogram: vitamin A (retinyl acetate), 20,000 IU; vitamin D3 (cholecalciferol), 6000 IU; vitamin E (dl-α-tocopheryl acetate), 80 IU; vitamin B1(thiamine monophosphate), 3 mg; vitamin B2 (riboflavin), 12 mg; vitamin B6 (pyridoxine hydrochloride), 8 mg; vitamin B12 (cyanocobalamin), 0.04 mg;, vitamin K3 (menadione), 4.8 mg; vitamin H (d biotin), 0.2 mg; vitamin PP (nicotinic acid), 48 mg; folic acid, 2 mg; calcium pantothenate, 20 mg; manganous oxide, 200 mg; ferrous carbonate, 80 mg; cupric sulphate pentahydrate, 20 mg; zinc oxide, 120 mg; basic carbonate monohydrate, 0.4 mg; anhydrous calcium iodate, 2 mg; sodium selenite, 0.4 mg; choline chloride, 800 mg; 4–6-phytase, 1800 FYT; D.L. methionine, 2600 mg; canthaxanthin, 8 mg.

**Table 2 animals-10-00081-t002:** Egg weight (g), egg components weight (g) and percentage (% on egg weight) and eggshell thickness (µm) as affected by the diet (C, HI25, HI50) and hen ages (20, 27, and 35 weeks).

Item	Diet, D			Hen Age, A			D	A	D × A	RMSE
	C	HI25	HI50	A20	A27	A35				
Egg Weight, g	57.56 ^b^	61.10 ^a^	60.82 ^a^	60.79	59.36	59.43	*	ns	ns	4.413
Eggshell, g	7.10 ^b^	7.65 ^a^	7.24 ^a,b^	7.26	7.60	7.14	*	ns	ns	0.643
Yolk, g	12.93 ^b^	14.30 ^a^	14.34 ^a^	13.92	13.46	14.19	***	ns	***	1.298
Albumen, g	37.53	39.15	38.47	39.62	37.43	38.10	ns	ns	ns	4.091
Eggshell, %	12.34 ^a^	12.54 ^a^	11.93 ^b^	11.96 ^Y^	12.83 ^X^	12.02 ^Y^	*	***	ns	0.741
Yolk, %	22.49	23.44	23.59	22.92	22.71	23.89	ns	ns	ns	1.896
Albumen, %	65.18	64.02	63.21	65.12	63.19	64.10	ns	ns	ns	4.016
Eggshell Thickness, µm	542 ^a^	516 ^a,b^	476 ^b^	499	540	494	*	ns	ns	72.249

RMSE: Root Means Square Error. a, b are significant different means among diets. X, Y are significant different means among hen ages. ns: not significant (*p* > 0.05), * *p* < 0.05, *** *p* < 0.001.

**Table 3 animals-10-00081-t003:** Color parameter (lightness, L*, redness, a*, and yellowness, b*, indexes) values of the yolk from hens fed three different diets (C, HI25, HI50) and collected at three different hen ages (20, 27, and 35 weeks).

Colour Parameter	Diet	Hen Age (Weeks)		
		A20	A27	A35
L*	C	65.804 ^ab^	64.686 ^b,Y^	67.865 ^a,XY^
	H25	66.665	68.553 ^X^	68.671 ^X^
	H50	68.858 ^a^	68.723 ^a,X^	65.537 ^b,Y^
a*	C	6.514 ^ab,X^	8.338 ^a,X^	2.874 ^b,X^
	H25	1.353 ^ab,Y^	2.395 ^a,Y^	−1.747 ^b,Y^
	H50	−0.437 ^b,Y^	−2.422 ^b,Z^	5.268 ^a,X^
b*	C	31.907 ^X^	32.428	32.558
	H25	33.463 ^X^	34.551	31.020
	H50	20.458 ^b,Y^	28.391 ^a^	33.300 ^a^

^a,b^: different superscript letters are significantly different means within the row. ^X,Y,Z^: different superscript letters are significantly different means within the column.

**Table 4 animals-10-00081-t004:** Technological properties (foam capacity and foam volume stability as %) and chemical composition (g/100 g of albumen) of the albumen from the eggs laid by hens fed three different diets (C, HI25, HI50) and collected at three different hen ages (20, 27, and 35 weeks).

Item	Diet, D			Hen Age, A			D	A	D × A	RMSE
	C	HI25	HI50	A20	A27	A35				
pH	8.12	8.08	8.05	8.19 ^X^	8.04 ^Y^	8.03 ^Y^	Ns	**	ns	0.143
Foam Capacity	43.33 ^a^	29.44 ^b^	20.74 ^b^	34.82 ^X^	39.44 ^X^	19.26 ^Y^	***	***	*	14.036
Foam Stability	59.20	52.55	63.05	52.58	57.34	64.89	Ns	ns	ns	19.331
Water Content	87.30	87.47	87.11	87.39 ^XY^	85.44 ^Y^	89.05 ^X^	Ns	***	ns	3.003
Crude Protein Content	11.20	11.03	11.39	11.11 ^X^	13.06 ^X^	9.45 ^Y^	Ns	***	ns	3.003

RMSE: Root Means Square Error. ^a,b^ are significant different means among diets. ^X,Y^ are significant different means among hen ages. ns: not significant (*p* > 0.05). * *p* < 0.05, ** *p* < 0.01, *** *p* < 0.001.

**Table 5 animals-10-00081-t005:** Physical and chemical characteristics of the angel cakes obtained by the albumen from the eggs laid by hens fed three different diets (C, HI25, HI50) and collected at three different hen ages (20, 27, and 35 weeks).

Item	Diet, D			Hen Age, A			D	A	D × A	RMSE
	C	HI25	HI50	A20	A27	A35				
Height, mm	20.25 ^a^	15.95 ^b^	18.65 ^a^	17.58	19.31	17.97	***	ns	*	2.39
Baking Loss, %	43.28 ^a^	44.67 ^a^	35.19 ^b^	41.28 ^X^	45.23 ^X^	36.63 ^Y^	***	***	*	7.871
*Colour Parameters*										
L*	87.956	87.653	88.360	88.257	87.690	88.023	ns	ns	ns	1.509
a*	−2.705	−2.742	−2.644	−2.669	−2.650	−2.772	ns	ns	ns	0.630
b*	13.494 ^a^	13.335 ^a^	11.936 ^b^	13.034	13.024	12.707	***	ns	***	0.847
*Texture*										
Hardness, N	6.39	5.52	5.95	5.90	6.85	5.13	ns	ns	ns	3.088
Cohesiveness	0.71	0.72	0.70	0.71	0.70	0.72	ns	ns	ns	0.027
Springiness	8.92	9.22	8.67	9.94 ^X^	7.89 ^Y^	8.98 ^X^	ns	***	ns	1.565
Chewiness	36.63	34.23	34.29	40.64 ^X^	36.42 ^X^	28.01^Y^	ns	***	ns	11.121

RMSE: Root Means Square Error. a, b are significant different means among diets. X, Y are significant different means among hen ages. ns: not significant (*p* > 0.05). * *p* < 0.05, *** *p* < 0.001.

**Table 6 animals-10-00081-t006:** Results of the interaction between the treatments (Diet × Hen Age) for angel cake characteristics.

Item	Diet	Hen Age (Weeks)		
		A20	A27	A35
Height, mm	C	19.45 ^X^	21.41 ^X^	19.88 ^X^
	H25	17.11 ^a,XY^	16.49 ^ab,Y^	14.26 ^b,Y^
	H50	16.17 ^b,Y^	20.03 ^a,X^	19.77 ^a,X^
Baking Loss, %	C	44.97	42.26 ^Y^	42.60 ^X^
	H25	40.86 ^b^	54.30 ^a,X^	38.84 ^b,X^
	H50	38.01 ^a^	39.12 ^a,Y^	28.44 ^b,Y^
b* Value	C	13.59 ^X^	13.78 ^X^	13.113 ^X^
	H25	12.584 ^b,Y^	13.49 ^ab,X^	13.932 ^a,X^
	H50	12.929 ^a,XY^	11.803 ^b,Y^	11.077 ^b,Y^

^a,b^ are significant different means within the row. ^X, Y^ are significant different means within the column.

**Table 7 animals-10-00081-t007:** Chemical composition (g/100 g fresh sample) of the angel cake as raw and baked batter.

Item	Diet, D			Hen Age, A			D	A	D × A	RMSE
	C	HI25	HI50	A20	A27	A35				
*Angel Cake Batter*										
Water	40.75	40.68	40.52	40.79 ^X^	39.74 ^Y^	41.42 ^X^	ns	***	ns	1.254
Crude protein	8.11	8.07	8.36	8.38	8.22	7.95	ns	ns	ns	0.744
*Angel Cake Baked*										
Water	28.61	29.13	27.72	29.68 ^X^	26.08 ^Y^	29.70 ^X^	ns	***	ns	2.736
Crude protein	9.77	9.65	10.16	9.96	10.08	9.54	ns	ns	ns	0.963

RMSE: Root Means Square Error. ^a,b^ are significant different means among diets. ^X,Y^ are significant different means among hen ages. ns: not significant (*p* > 0.05), *** *p* < 0.001.

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
