# Peer review of "Quality of Eggs and Albumen Technological Properties as Affected by *Hermetia Illucens* Larvae Meal in Hens’ Diet and Hen Age"

_animals, 2020, doi:10.3390/ani10010081_

Round 1

Reviewer 1 Report

Manuscript revision ID: animals-641004titled:

Quality of eggs and albumen technological properties 3 as affected by Hermetia illucens larvae meal in hens’ 4 diet and deposition week

Abstact

Please write again, provide the most important results and conclusions, including applications. Are the mixtures also iso-energetic??

Introduction

Rewrite again. Please, highlight the aspect of nutrition and the possibility of using insect meal, and then describe the technological aspect (shorten). It should be noted that currently there is no legal basis for feeding poultry with insect meal. The law applies (EC Regulation No. 1069/2009) prohibiting the use of insect poppies in feeding livestock, as well as that these meals do not meet quality standards, because there are no ...

Material and methods

Please provide the approval number for animal testing.

In how many replicates were the analyzes and measurements carried out?

L144 - please complete the results with statistical parameters characterizing the variability in the group, such as SD or SEM.

Discussion

Please strengthen the discussion by quoting nutritional studies conducted under similar conditions and using Hermetia illucens larvae meal. Comparative data is missing.

L254-255 - iso-protein and isoenergetic diets are necessary to form the correct methodical structure of the experiment, but do not guarantee the identical biological value of the experimental mixes protein. This sentence is false and indicates a lack of nutritional knowledge of the authors. Please remove it.

L262- very bold statement ... Poorly supported by literature, LIZ is AA very well known in the light of poultry nutritional requirements. Pay attention to the other amino acids, please.

L321-331 - the culinary and technological part should be limited, due to the nature of the Journal.

Conclusion

Please write again. It doesn't sum up the research, it's too laconic.

L333- the use of unstuffed feed for poultry production cannot be recommended. Please provide application forms.

Reviewer 2 Report

The submitted manuscripts evaluates egg, albumen and product quality of hens fed by two levels of Hermetia illucens larvae meal in feed mixtures. The topic is very interesting and brings some new results. However, some data do not reflect common analyses using in a feeding experiment. Authors also use very specific terms which are not common in this type of analyses. Data of egg quality have been received on frozen eggs which is not common and results can be affected by this method. Data also analyze very small number of eggs. Therefore, I have doubts if the paper gives reliable results.

Comments:

L 16: eggs were sampled 1st, 7th and 15th week but is not clear of what, age, experiment? In the whole manuscript, incidence of the shell, yolk, albumen – this is not correct term and should be replaced by shell percentage and etc. Similarly, week of deposition is not clear what does it mean? Deposition of what? Presumably, the term is a week of the experiment Also, age of hens during the experiment is not clear, in the material is necessary to write from which week of age of hens the experiment started and which finished To analyze 6 eggs from each group and age is not sufficient to get reliable results Authors analyzed frozen eggs but there is not explain why eggs were not analyzed fresh. To analyze frozen eggs is not common and is not possible to compare the results with fresh eggs, which is in Discussion. Egg analyses are poorly described Feeding larvae meal needs to include a nutritional composition of the meal Table 1 metabolizable energy in Europe is expressed by joules not kcal, feed color evaluation is not important if authors did not evaluate hen`s behavior Table 2 does not give reliable results due to a small number of analyzed eggs Table 3 is not clearly designed; the design should have been similar to T2 and T4 Table 4 Row protein is not correct, correct is crude protein Figure 1 describes the same results which are in Table 4 L 254: the sentence is not correct because egg weight is mainly affected by metabolizable energy L 259-264: lysine has a minor effect on egg weight, egg weight is affected by methionine. Therefore, explanation of the results is not correct L 281: is not clear what is egg deposition L 285-290 comparison of the results of frozen and fresh eggs is not correct

Reviewer 3 Report

Dear Authors, the manuscript entitled "Quality of eggs and albumen technological properties as affected by Hermetia illucens larvae meal in hens' diet and deposition week" is interesting and may be a good source of knowledge regarding the use of insects in the feeding of laying hens .

The manuscript requires a slight correction. After correction could be recommend for the publication processing.

Simple Summary: it is written as the Abstract. How it will be valuable to society? Try to improve if it is possibility;

Abstract: lack of conlusions;

Introduction. Generally, it is good introduction, but if authors could write little more about the soybean meal (protein content etc.) and about the factor that is main subject of this research (larva meal).

line 42-43: "... eggs are the second fastest growing protein source in the world, with more than 50% growth forecast in the next 2 decades ..." - maybe some numbers should be add?

line 45: ... "However, around 30% ..." - 30% of what? 100 eggs? We have no idea how many eggs is in 30%.

Materials and Methods - could be described more

2.1. In vivo trails

more information about the conditions of laying hens keeping (lightening program

Tabel 1. In metabolizable energy you added units (kcal / kg) in table. Why you did not add units in the other items?

"shell" e.i. in table 2 and in 154 line. This word is more like shell from the sea etc. "eggshell" is better.

Table 3. No significant differences between 6.514 and 2.874 in L * parameter of C group? The same: b *, H50 group: 7th week and 15th week?

Results: it is written corectly.

Discussion: it is written corectly

Conclusion: "The use of eggs laid by hens fed the H50 diet ..." maybe "The obtained eggs from laying hens fed the H50 diet ...?

Check the section References regarding to the instruction for Authors.

Reviewer 4 Report

Reviewing the manuscript animals 641004

Quality of eggs and albumen technological properties as affected by Hermetia illucens larvae meal in hens’ diet and deposition week

General comments:

The manuscript is mostly well very written and the content is novel and highly relevant to the poultry industry and meets the scope of the journal.

Crucial information to decide about the suitability for publication is missing. The major limitation is that there is no information disclosed about the amino acid composition of the diets. This is absolutely crucial for animal performance and interpretation of the results. Was the Amino Acid composition balanced? Were all essential amino acids administered in sufficient quantities and at the adequate ratio? This can affect the albumen quality significantly and as such an imbalance of the test diets may be responsible for the results observed rather than the fact the HI was used. Similarly, information about crude ash, calcium, phosphorus and other minerals is lacking which are crucial for egg shell quality. Depending on the development stage used, HI can include up to 5% calcium which would impact egg shell quality. Despite it being stated, the diets appear not to be balanced (and the discussion mentions this), which results in a methological error, the conclusion is therefore not justified and the research should not be considered for publication. Egg size varied significantly and this is a major factor that impacts egg shell quality. The current discussion is therefore off-topic, ignoring this information widely but rather trying to explain the observed findings based on the treatments. Furthermore, results are discussed (chemical composition of albumen) that are not presented.

Summary:

Please clarify if the diet was still balanced and still met the breeder’s nutrient recommendations.

Line 18: “affect the chemical composition of the albumen”- only details about physical albumen behaviour is presented in eth manuscript, note the chemical composition. Please include chemical composition details or delete/change this sentence.

Line 23: how do you define “shell incidence”? unclear what parameter is meant by this. 

Please outline the consequences of the impact of HI on albumen foaming.

Please include information about how many hens and the number of treatment replicates that were used for this experiment, at what age and what breed and also if this research was performed on a commercial farm or university (small scale) setting involving manual handling. Was the housing caged, barn or free range?

Please clarify if the HI meal was full fat or defatted.

Abstract:

Line 31: please define “shell incidence”. Are you referring to cracked eggs?

Please include information about how many hens and the number of treatment replicates that were used for this experiment, at what age and what breed and also if this research was performed on a commercial farm or university (small scale) setting involving manual handling. Was the housing caged, barn or free range? Was the diet isonutrigenous?

Line 35: “angel cake height” – please include information that angel cake height was evaluated using a standardized and validated method.  

Please provide information about the statistical method used and how chewiness was evaluated. Please provide information about all the parameters evaluated.

Please clarify if the HI meal was full fat or defatted.

Introduction:

Line 42: “Today, eggs are the second fastest growing protein” please refer to the year rather than “today”, and also rephrase “fastest growing” to “the fastest growing industry” or similar, since eggs are not growing but rather being produced by the hen and this still takes 25 hours/egg.

Line 45-46: “However, around 30% of the eggs produced are processed in the egg industry”- this depends strongly on the country., some countries use only 10 % or less as egg product. Please specify which country/countries you are referring to.

Line 51: “ missing at the end of the quote.

Please provide a hypothesis why HI would impact egg quality parameters and provide background what other components affect the technological egg properties.

Materials and Methods:

Line 72: “Amtsgericht Potsdam” meant “courthouse Potsdam” and is unlikely the city and country where the Hermetia company is located. Please include

Line 74: please replace “hosted” with “housed”

Line 75: please specify if the private laying hen farm is producing eggs commercially and what the total flock size was.

Line 81: how was the diets manually administrated? Was each hen individually fed? In what type of feeder? Ad libitum? Or how many times a day? Was the diet mash, crumbs or pelleted? This is important as this may or may not allow hens to select diet components.

Line 80: “in order to result isoproteic, isolipidic and isoenergetic”. Table 1 shows very different values for Ether extract which does not support the claim thatt the diets were isolipidic. Amino acid composition is absolutely crucial for animal performance and as such information about the amino acid content (was it balanced? Were all essential amino acids administered in sufficient quantities and at the adequate ratio?

Line 81: in the modified cages, how and how often were the eggs collected? This is important as next box management impacts egg quality. Was there a nest box at all?

Line 84: “6 of them per each group in 3 different weeks” – unclear. Did you analyse 6 eggs/replicate cage? I assume that the values of the 6 eggs were averaged since each cage was considered as one statistical unit? Please clarify. During the week of analysis, were eggs collected on 7 consecutive days and averaged or only at one day?

Please provide information about the HI larvae stage used, how the meal was processed (heat treatment? Grinding method?) and how the feed was offered (mash, crumbs, pellet?) since this all affects amino acid availability. In the abstract you mention “partially defatted”- please provide details here- how much fat, what method was used to extract the fat?

Table 1:

Please include the information about the Amino acid composition of the diets. This can affect the albumen quality significantly and as such an imbalance of the test diets may be responsible for the results observed rather than the fact the HI was used. Similarly, please include information about crude ash, calcium, phosphorus and other minerals as calculated and analysed since this is crucial for egg shell quality. Depending on the development stage used, HI can include up to 5% calcium which would impact egg shell quality.

Units are missing

Adding the quantities of ingredients of the HI50 does results in 991 (kg?). I am wondering what the other ingredients are so the total volume is 100% or 1000 kg.

Quantities of HI25 results in 1005 (kg?). if these quantities were truly used then the diets are not comparable since the hens would need to vary their intake to ingest the same amount of energy. Please double check your numbers or remove the statement “isoenergetic”.

Change “chemical-nutritional composition” to: “chemical composition”

How were colour values determined? What yolk colour number was targeted?

Please include information about crude ash, calcium, phosphorus and other minerals as calculated and analysed since this is crucial for egg shell quality. Depending on the development stage used, HI can include up to 5% calcium which would impact egg shell quality.

Line 100: it is nowhere mentioned that eggs were frozen, why do they need to be thawn?

Line 101: how doid you calculate the “relative incidence”? are you referring to relative egg shell weight, relative albumen weight and relative yolk weight in %? Did you determine egg shell weight with or without inner membrane and also was the egg shell dried (for how long?) ? If not, albumen remnants would have altered the results resulting in large variation and incorrect results. Please clarify.

Line 102: please provide the model.

Line 103: which points? Equator, round end or apex? With or without inner membrane? After drying or before?

Line 104, 108 and all other relevant: please provide information on the model used.

Line 106: was it manual stirring or what machine was used at which setting?

Line 110-115: it is unclear is the 6 eggs obtained from each cage were pooled before evaluated or evaluated one-by one and then averaged or if only one egg / replicate was investigated. It is also unclear if 1 egg / week was investigated or if eggs were evaluated for 7 consecutive days. Please clarify.

Line 128: please provide details about model, manufacturer, city, county of the silicone pan used.

Line 129: please replace “cooked” with “baked” and refer to “baking loss (line 131 and others).

Line 129: please provide details about model, manufacturer, city, county of the static oven used.

Line 143: please provide the formulas used.

Line 147: please include information about the city and county of the SAS distributor as well as the version used. Were outliers tested? Was normality tested? How was data handled that was not normally distributed and which data was not normally distributed?

Line 144 ff: the analysis should include regression analysis to evaluate a dose response.

Please state the animal ethics approval authority number and authority that approved the study.

Results:

Table 2:

Units missing. Please check all tables.

Please change “incidence” to “relative shell weight” etc

Please apply 3-digit rule to the results reported (xx.x, x.xx, 0.xxx). no reason to report 3 digits after the decimal point for shell weight, for example.

Please explain the abbreviation RMSE – please apply to all tables

Please refer to the actual p-values rather than stars

Line 161: numerical differences are irrelevant and should not be mentioned.

Discussion:

Line 254-255: there is no evidence that diets were balanced with amino acids which significantly affects albumen quality. Based on the current numbers provided in table 1 diets were also not isoenergetic since the diet composition provided did not add up to 1000 (kg?) or 100%.

The statement is therefore not correct and should be deleted.

Lines 256 ff: egg weight also changes based on body weight. Body weight of hens at the end of the trial needs to be recorded to exclude this factor and allow for the conclusion currently stated.

Lines 260-261: was the lysine (and other essential amino acids) requirements met according to breeder guidelines in the control diet???

Line 261- 264: please review the impact of Amino acids on egg size & quality as outlined by Leeson & Summers, poultry nutrition and re-write this section.

Line 268-269: “Hence, the production of heavier eggs in the HI50 group might explain 269 the significant reduction in shell incidence and even in its thickness” – this is indeed most likely the case, the impact you see on egg quality is not because of the HI inclusion (what would be your hypothesis here) but because of the increasing egg size. It is absolutely crucial that you provide information on diet calcium and phosphate content to may ANY conclusion and you also need to explain why egg size was larger – with considering hen body weight as the major factor.

Line 273: “egg internal quality is commonly associated to the yolk quality” egg internals quality is also commonly associated with albumen height, Haugh Unit,

Line 298-299: “Nevertheless, these last two proteins contributed to the final volume of the Angel cakes made with them…” the results on chemical albumin composition and nowhere described, neither in the methods nor in results section and therefore cannot be discussed here in the current form. Please refer clearly to the literature.

Line 304: “The albumen chemical composition varied along the weeks” – as above.

Reviewing the manuscript animals 641004

Quality of eggs and albumen technological properties as affected by Hermetia illucens larvae meal in hens’ diet and deposition week

General comments:

The manuscript is mostly well very written and the content is novel and highly relevant to the poultry industry and meets the scope of the journal.

Crucial information to decide about the suitability for publication is missing. The major limitation is that there is no information disclosed about the amino acid composition of the diets. This is absolutely crucial for animal performance and interpretation of the results. Was the Amino Acid composition balanced? Were all essential amino acids administered in sufficient quantities and at the adequate ratio? This can affect the albumen quality significantly and as such an imbalance of the test diets may be responsible for the results observed rather than the fact the HI was used. Similarly, information about crude ash, calcium, phosphorus and other minerals is lacking which are crucial for egg shell quality. Depending on the development stage used, HI can include up to 5% calcium which would impact egg shell quality. Despite it being stated, the diets appear not to be balanced (and the discussion mentions this), which results in a methological error, the conclusion is therefore not justified and the research should not be considered for publication. Egg size varied significantly and this is a major factor that impacts egg shell quality. The current discussion is therefore off-topic, ignoring this information widely but rather trying to explain the observed findings based on the treatments. Furthermore, results are discussed (chemical composition of albumen) that are not presented.

Summary:

Please clarify if the diet was still balanced and still met the breeder’s nutrient recommendations.

Line 18: “affect the chemical composition of the albumen”- only details about physical albumen behaviour is presented in eth manuscript, note the chemical composition. Please include chemical composition details or delete/change this sentence.

Line 23: how do you define “shell incidence”? unclear what parameter is meant by this. 

Please outline the consequences of the impact of HI on albumen foaming.

Please include information about how many hens and the number of treatment replicates that were used for this experiment, at what age and what breed and also if this research was performed on a commercial farm or university (small scale) setting involving manual handling. Was the housing caged, barn or free range?

Please clarify if the HI meal was full fat or defatted.

Abstract:

Line 31: please define “shell incidence”. Are you referring to cracked eggs?

Please include information about how many hens and the number of treatment replicates that were used for this experiment, at what age and what breed and also if this research was performed on a commercial farm or university (small scale) setting involving manual handling. Was the housing caged, barn or free range? Was the diet isonutrigenous?

Line 35: “angel cake height” – please include information that angel cake height was evaluated using a standardized and validated method.  

Please provide information about the statistical method used and how chewiness was evaluated. Please provide information about all the parameters evaluated.

Please clarify if the HI meal was full fat or defatted.

Introduction:

Line 42: “Today, eggs are the second fastest growing protein” please refer to the year rather than “today”, and also rephrase “fastest growing” to “the fastest growing industry” or similar, since eggs are not growing but rather being produced by the hen and this still takes 25 hours/egg.

Line 45-46: “However, around 30% of the eggs produced are processed in the egg industry”- this depends strongly on the country., some countries use only 10 % or less as egg product. Please specify which country/countries you are referring to.

Line 51: “ missing at the end of the quote.

Please provide a hypothesis why HI would impact egg quality parameters and provide background what other components affect the technological egg properties.

Materials and Methods:

Line 72: “Amtsgericht Potsdam” meant “courthouse Potsdam” and is unlikely the city and country where the Hermetia company is located. Please include

Line 74: please replace “hosted” with “housed”

Line 75: please specify if the private laying hen farm is producing eggs commercially and what the total flock size was.

Line 81: how was the diets manually administrated? Was each hen individually fed? In what type of feeder? Ad libitum? Or how many times a day? Was the diet mash, crumbs or pelleted? This is important as this may or may not allow hens to select diet components.

Line 80: “in order to result isoproteic, isolipidic and isoenergetic”. Table 1 shows very different values for Ether extract which does not support the claim thatt the diets were isolipidic. Amino acid composition is absolutely crucial for animal performance and as such information about the amino acid content (was it balanced? Were all essential amino acids administered in sufficient quantities and at the adequate ratio?

Line 81: in the modified cages, how and how often were the eggs collected? This is important as next box management impacts egg quality. Was there a nest box at all?

Line 84: “6 of them per each group in 3 different weeks” – unclear. Did you analyse 6 eggs/replicate cage? I assume that the values of the 6 eggs were averaged since each cage was considered as one statistical unit? Please clarify. During the week of analysis, were eggs collected on 7 consecutive days and averaged or only at one day?

Please provide information about the HI larvae stage used, how the meal was processed (heat treatment? Grinding method?) and how the feed was offered (mash, crumbs, pellet?) since this all affects amino acid availability. In the abstract you mention “partially defatted”- please provide details here- how much fat, what method was used to extract the fat?

Table 1:

Please include the information about the Amino acid composition of the diets. This can affect the albumen quality significantly and as such an imbalance of the test diets may be responsible for the results observed rather than the fact the HI was used. Similarly, please include information about crude ash, calcium, phosphorus and other minerals as calculated and analysed since this is crucial for egg shell quality. Depending on the development stage used, HI can include up to 5% calcium which would impact egg shell quality.

Units are missing

Adding the quantities of ingredients of the HI50 does results in 991 (kg?). I am wondering what the other ingredients are so the total volume is 100% or 1000 kg.

Quantities of HI25 results in 1005 (kg?). if these quantities were truly used then the diets are not comparable since the hens would need to vary their intake to ingest the same amount of energy. Please double check your numbers or remove the statement “isoenergetic”.

Change “chemical-nutritional composition” to: “chemical composition”

How were colour values determined? What yolk colour number was targeted?

Please include information about crude ash, calcium, phosphorus and other minerals as calculated and analysed since this is crucial for egg shell quality. Depending on the development stage used, HI can include up to 5% calcium which would impact egg shell quality.

Line 100: it is nowhere mentioned that eggs were frozen, why do they need to be thawn?

Line 101: how doid you calculate the “relative incidence”? are you referring to relative egg shell weight, relative albumen weight and relative yolk weight in %? Did you determine egg shell weight with or without inner membrane and also was the egg shell dried (for how long?) ? If not, albumen remnants would have altered the results resulting in large variation and incorrect results. Please clarify.

Line 102: please provide the model.

Line 103: which points? Equator, round end or apex? With or without inner membrane? After drying or before?

Line 104, 108 and all other relevant: please provide information on the model used.

Line 106: was it manual stirring or what machine was used at which setting?

Line 110-115: it is unclear is the 6 eggs obtained from each cage were pooled before evaluated or evaluated one-by one and then averaged or if only one egg / replicate was investigated. It is also unclear if 1 egg / week was investigated or if eggs were evaluated for 7 consecutive days. Please clarify.

Line 128: please provide details about model, manufacturer, city, county of the silicone pan used.

Line 129: please replace “cooked” with “baked” and refer to “baking loss (line 131 and others).

Line 129: please provide details about model, manufacturer, city, county of the static oven used.

Line 143: please provide the formulas used.

Line 147: please include information about the city and county of the SAS distributor as well as the version used. Were outliers tested? Was normality tested? How was data handled that was not normally distributed and which data was not normally distributed?

Line 144 ff: the analysis should include regression analysis to evaluate a dose response.

Please state the animal ethics approval authority number and authority that approved the study.

Results:

Table 2:

Units missing. Please check all tables.

Please change “incidence” to “relative shell weight” etc

Please apply 3-digit rule to the results reported (xx.x, x.xx, 0.xxx). no reason to report 3 digits after the decimal point for shell weight, for example.

Please explain the abbreviation RMSE – please apply to all tables

Please refer to the actual p-values rather than stars

Line 161: numerical differences are irrelevant and should not be mentioned.

Discussion:

Line 254-255: there is no evidence that diets were balanced with amino acids which significantly affects albumen quality. Based on the current numbers provided in table 1 diets were also not isoenergetic since the diet composition provided did not add up to 1000 (kg?) or 100%.

The statement is therefore not correct and should be deleted.

Lines 256 ff: egg weight also changes based on body weight. Body weight of hens at the end of the trial needs to be recorded to exclude this factor and allow for the conclusion currently stated.

Lines 260-261: was the lysine (and other essential amino acids) requirements met according to breeder guidelines in the control diet???

Line 261- 264: please review the impact of Amino acids on egg size & quality as outlined by Leeson & Summers, poultry nutrition and re-write this section.

Line 268-269: “Hence, the production of heavier eggs in the HI50 group might explain 269 the significant reduction in shell incidence and even in its thickness” – this is indeed most likely the case, the impact you see on egg quality is not because of the HI inclusion (what would be your hypothesis here) but because of the increasing egg size. It is absolutely crucial that you provide information on diet calcium and phosphate content to may ANY conclusion and you also need to explain why egg size was larger – with considering hen body weight as the major factor.

Line 273: “egg internal quality is commonly associated to the yolk quality” egg internals quality is also commonly associated with albumen height, Haugh Unit,

Line 298-299: “Nevertheless, these last two proteins contributed to the final volume of the Angel cakes made with them…” the results on chemical albumin composition and nowhere described, neither in the methods nor in results section and therefore cannot be discussed here in the current form. Please refer clearly to the literature.

Line 304: “The albumen chemical composition varied along the weeks” – as above.

Round 2

Reviewer 2 Report

Authors corrected original manuscript but still the revised form contains parts, which were considered as main to reject. The firs, t is the number of analysed eggs. Even that authors increased the number of eggs to 10 per group and age, still the number is very low to get reliable results because of the individual egg and age egg variability. The second, is the analysis of the frozen and then thawed eggs because both processes may affect internal and external egg quality. In addition, to compare these results with results on fresh eggs in Discussion is not correct. For missing composition of Hermetia illucens larvae meal and citation Bovera et al. (2018) is acceptable only in a case that the results are from one experiment. For two main reasons given above I do not recommend publishing the manuscript in Animals.